# Postnatal cytomegalovirus infection and its effect on hearing and neurodevelopmental outcomes among infants aged 3–10 months: A cohort study in Eastern Uganda

Noela Regina Akwi Okalany[1]*, David Mukunya[2‡], Peter Olupot-Olupot[2,3‡], Martin Chebet[1,4‡], Francis Okello[1,2‡], Andrew D. Weeks[5‡], Fred Bisso[6‡], Thorkild Tylleskär[1‡], Kathy Burgoine[1,3,6☉], Ingunn Marie Stadskleiv Engebretsen[1☉]

1 Department of Global Public Health and Primary Care, Centre for International Health, University of Bergen, Bergen, Norway, 2 Department of Community and Public Health, Busitema University, Mbale, Uganda, 3 Mbale Clinical Research Institute, Mbale, Uganda, 4 Department of Paediatrics and Child Health, Busitema University, Mbale, Uganda, 5 Department of Women's and Children's Health, Sanyu Research Unit, University of Liverpool, Liverpool, United Kingdom, 6 Mbale Regional Referral Hospital, Mbale, Uganda

☉ These authors contributed equally to this work.
‡ These authors also contributed equally to this work.
* n.okalany@gmail.com

**Data Availability Statement:** Additionally, with regards to data availability, this study was derived

## Abstract

### Background

Hearing impairment and neurodevelopmental disorders pose a significant global health burden in children. The link between postnatal cytomegalovirus (CMV) infection and these outcomes remains unclear. This study explored the association of postnatal CMV infection with hearing and neurodevelopmental outcomes in term infants aged 3 to 10 months.

### Methods

This was a cohort sub-study within the BabyGel cluster randomised trial in Eastern Uganda. From 1265 term infants screened for CMV, 219 were negative at birth but positive at 3 months, and were age-matched with 219 CMV-negative controls. CMV status was determined by PCR screening of saliva samples, with positive results confirmed using urine samples (Chai Open qPCR, Santa Clara, CA). From the established cohort, 424 infants were successfully followed up between 3 to 10 months of age. Clinical assessments included neurodevelopmental evaluation using the Malawi Developmental Assessment Tool, the Hammersmith Infant Neurological Examination, and hearing screening using Otoacoustic Emission testing (Otoport Lite, Otodynamics Limited). Statistical analyses were performed using descriptive statistics, chi-square tests and log binomial regression models with Stata 18.

### Results

Of the 424 infants included in the study, 206 were postnatal CMV-infected and 218 were uninfected. Neurodevelopmental assessments indicated no differences between postnatal

from the BabyGel trial, and as a sub-study in a clinical trial, we acknowledge the importance of adhering to the guidance provided by the BabyGel Trial Management Group (TMG). We confirm that we are providing a minimal de-identified dataset on a recommended public repository containing the variables required to replicate the key outcomes reported in this paper (https://doi.org/10.6084/m9.figshare.28270673). As per the guidance of the BabyGel Trial Management Group (TMG), the full dataset will be made publicly available only following the publication of the main BabyGel trial paper. Until then, only the specific variables used in this study are being shared. For additional data requests, please contact the BabyGel TMG or Data Management and Access Review Group (DMARG) at aweeks@liverpool.ac.uk or brian.faragher@lstmed.ac.uk.

**Funding:** This research was part of the EDCTP2 programme supported by the European Union under grant number RIA2017MC-2029. The funders had no role in study design, data collection and analysis, decision to publish, or preparation of the manuscript. The content is solely the responsibility of the authors and does not necessarily represent the official views of EDCTP.

**Competing interests:** The authors have declared that no competing interests exist.

CMV-infected infants and uninfected groups (ARR 0.88, 95% CI [0.67, 1.15], p = 0.346). Hearing screening revealed a 1.99-fold increased risk of a positive result for postnatal CMV-infected infants compared to uninfected infants (67/106 vs. 39/106, 95% CI [1.27, 3.12], p = 0.003).

## Conclusion

Postnatal CMV infection was associated with more positive hearing screenings, though no significant differences in neurodevelopmental outcomes were observed in early infancy. Exploration into the feasibility of incorporating hearing and CMV screening into routine care will play a vital role in early identification and intervention, improving the management of both hearing and CMV-related conditions in resource-limited settings.

## Introduction

Congenital CMV is known to significantly contribute to sensorineural hearing and neurodevelopmental impairment in high-resource settings [1,2]. Emerging data suggest that the disease burden and disability associated with CMV are even higher in resource-limited settings [3]. Postnatal cytomegalovirus (postnatal CMV) is a viral infection that occurs after birth, particularly impacting preterm infants and those with very low birth weight (VLBW, <1500g) due to their relatively immature immune systems [4,5]. This can lead to long-term morbidity, including hearing and neurodevelopmental impairment [6,7].

Transmission of postnatal CMV can occur via various routes, including breast milk, blood transfusions, and direct contact with bodily fluids from CMV-excreting contacts [8,9]. Breast milk is the main transmission route, especially in populations with high viral seroprevalence [10]. Due to postpartum viral reactivation, many lactating mothers shed CMV DNA, with peak shedding around one month after delivery [11]. Despite the presence of maternal antibodies, like immunoglobulin G, infants often become infected with CMV before one year of age [12]. Sub-Saharan Africa has a high CMV population seroprevalence of 90% or higher [13–15]. The region also contends with a substantial burden of other infectious and endemic illnesses, such as HIV/AIDS and malaria, which can complicate the immune response in infected individuals and impact viral shedding [16–20]. Furthermore, studies have shown that CMV transmission via breastmilk in these populations ranges from 58% to 76%, indicating a substantial risk of infection [21].

It has been postulated that postnatal CMV infection is generally benign in healthy full-term infants, but data are limited [22,23]. Understanding the effect of postnatal CMV infection in healthy term infants from populations with a high burden of both CMV and infectious disease is needed to guide clinical management and interventions. The aim of this study was to investigate the effect of postnatal CMV infection on hearing and neurodevelopmental outcomes in term infants.

## Methods

### Study design and setting

The study was a matched cohort sub-study nested within a larger prospective cohort study of congenital cytomegalovirus (congenital CMV) infection, which was part of a cluster-randomised controlled trial in Eastern Uganda, the BabyGel trial, which aimed to evaluate the effectiveness of household alcohol-based hand rub (ABHR) in preventing neonatal infections such

as sepsis, diarrhoea, and pneumonia by improving hygiene practices among postpartum mothers and their newborns, with follow-up of mothers and infants for three months post-delivery to monitor health outcomes [24]. This sub-study specifically focused on postnatal CMV infection and its impact on early hearing and neurodevelopmental outcomes in infants. It included postnatal CMV-infected infants who did not have congenital CMV (negative PCR CMV test within 21 days of birth) but later tested positive for CMV at three months of age. Biological sampling was conducted with saliva and urine samples which have demonstrated high accuracy, with saliva showing a sensitivity of over 97% and specificity of 99% [25], and urine achieving 100% sensitivity and 99% specificity [26]. Laboratory diagnostic methods using PCR testing are described (S1 Appendix). Age matched infants were those who tested negative at both 21 days and three months of age. All the matched pairs were born within ten days of each other. The study followed the infants between July 2023 and April 2024 in the Mbale and Budaka districts of Eastern Uganda.

## Sample size

The sample size was determined using a 1:1 ratio of postnatal CMV-infected to uninfected infants.

From 1265 term infants screened for CMV, 219 babies did not have CMV at birth but tested positive for CMV at 3 months of age. They were matched with 219 age-matched CMV negative controls, resulting in a total sample of 438 participants. Of these, 14 infants were excluded from the final analysis due to loss to follow-up (n = 10), missed assessments (n = 3), and death (n = 1). This resulted in a final sample of 424 infants with complete data for otoacoustic emission (OAE) testing, the Hammersmith Infant Neurological Examination (HINE), and the Malawi Developmental Assessment Tool (MDAT), comprising 206 postnatal CMV-infected and 218 CMV-uninfected infants (Fig 1).

## Study procedures

**Subject recruitment and consent procedures.** Participants in this sub-study were selected from mothers who were first enrolled in the BabyGel trial. Recruitment for the trial was conducted by village health team members (VHTs) and midwives from the community and antenatal clinics at local health centres. Potential participants were identified, visited at home, assessed for eligibility, and provided with a detailed explanation of the trial before giving informed consent.

Following delivery, mothers participating in the BabyGel trial were invited to join the CMV sub-study within the first week of their infant's life. Infants were eligible if they were born to these participants, delivered after 34 weeks of gestation, and resided with their parents or legal guardians in the participating villages.

The study enrolled and followed up participants between 29th July 2023 and 30th April 2024, during which written informed consent was obtained from the parents or legal guardians of all infant participants. Detailed explanations of the study's objectives, procedures, risks, and benefits were provided, ensuring that participation was entirely voluntary. The study adhered to ethical standards outlined in the Declaration of Helsinki.

**CMV testing and follow-up.** Congenital CMV infection was defined as a positive saliva sample within 21 days of birth, confirmed with urine polymerase chain reaction (PCR) testing. Infants who did not test positive for congenital CMV were followed up at three months of age. At this follow-up visit, identical samples were collected and tested for CMV. Infants testing positive were considered postnatal CMV-infected, while those testing negative were considered postnatal CMV-uninfected.

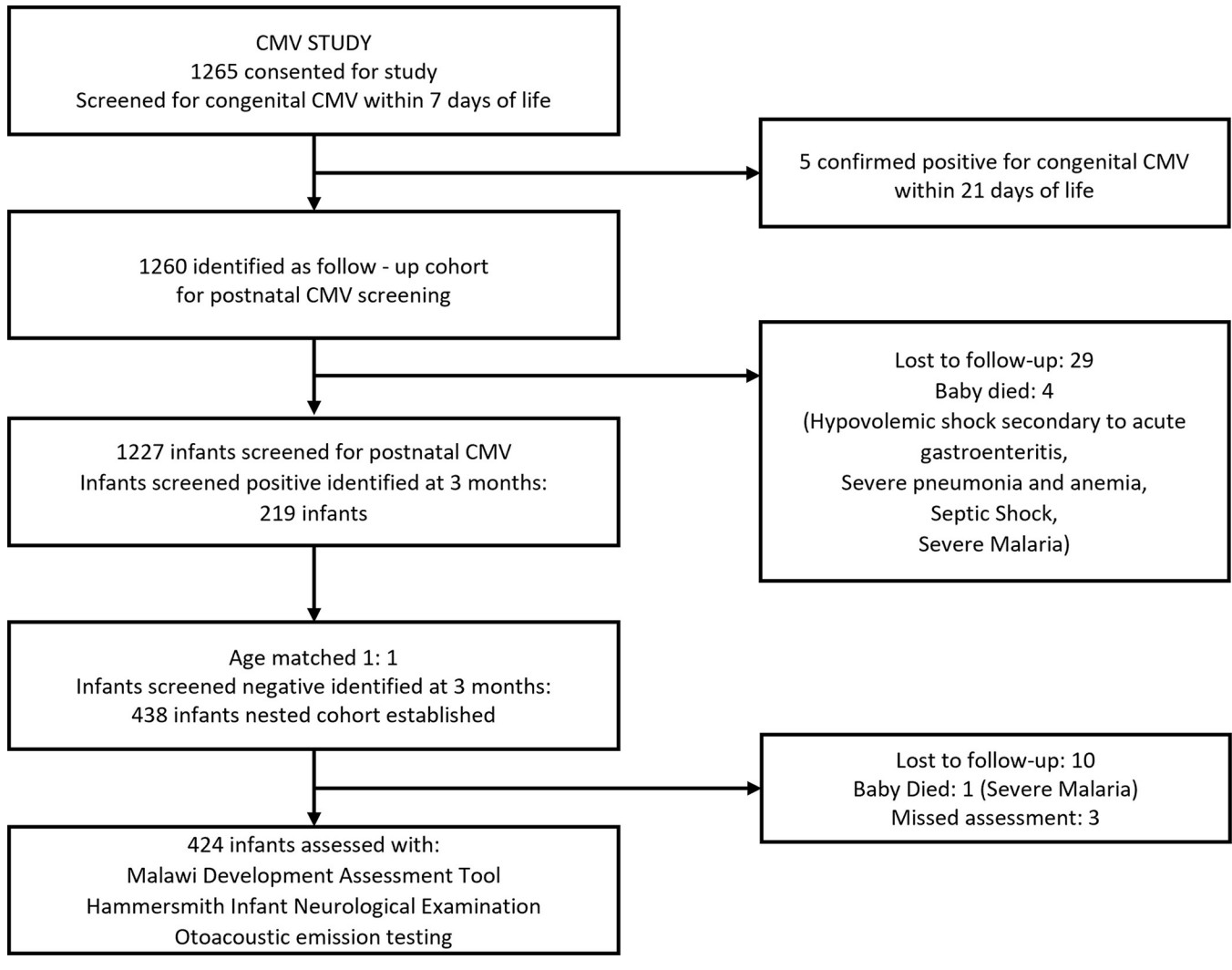

**Fig 1. Study flowchart of infant screening and follow-up for postnatal CMV.**

**Data collection.**    Data was collected through structured questionnaires comprising household and parental demographics, antenatal data, and infant data. Maternal data included age, parity, history of miscarriages, HIV status, malaria status, education level, and hand hygiene practices before baby contact. Household characteristics included residential setting, family meal-sharing practices, wealth index, availability of handwashing facilities and soap, access to improved drinking water, and sanitation practices. Infant-specific data comprised sex, birthweight, respiratory or infectious diagnoses, hospitalisation history, medication and antibiotic use, vaccination status, and exclusive breastfeeding practices.

**Clinical assessments.**    For clinical assessments, we evaluated neurodevelopment using the Malawi Development Assessment Tool (MDAT) [27], conducted a neurological evaluation using the Hammersmith Infant Neurological Examination (HINE) [28], and assessed hearing using Otoacoustic Emission testing. Details concerning the diagnostic methods and assessment tools are described in the prospective paper 'Congenital cytomegalovirus infection in Eastern Uganda' [29]. Five research assistants were trained to administer the neurodevelopmental assessment tools across the study sites. The principal investigator (NRAO) trained and

supervised the research assistants, who were midwives trained in data collection and documentation. For neurodevelopmental assessments, infants scoring below age-appropriate clinical assessment threshold levels for referral (-2 SD) were identified.

For hearing assessments, the principal investigator (NRAO) conducted the hearing screening, the presence of distortion product otoacoustic emissions (DPOAEs) was evaluated in both ears, and a positive screening result was given if DPOAEs were either not detected or measured below the threshold for normative values in one or both ears. Infants with abnormal neurodevelopmental assessment scores or a positive screening result from the hearing assessment were subsequently directed to Paediatric and Ear, Nose, and Throat (ENT) clinics for further evaluation. Additionally, caregivers were provided with health education to support improved outcomes.

**Statistical analysis.** Continuous variables which were normally distributed were reported as means with standard deviations; medians and interquartile ranges were used for those that were not normally distributed. Comparisons between the postnatal CMV-infected and uninfected groups regarding categorical early hearing outcomes, developmental outcomes (MDAT), and neurological outcomes (HINE) were conducted using cross-tabulations and chi-squared tests. The outcomes included hearing status (normal vs. positive), development-for-age Z-scores (DAZ) groups ranging from -3 to <-2, -2 to <-1, -1 to <0, 0 to <1 and 1 to <2, and neurological status (above vs. below age-specific thresholds).

To further compare neurodevelopmental outcomes between postnatal CMV-infected and uninfected groups, the mean (95% Confidence Interval (CI)) of MDAT DAZ scores across various domains (gross motor, fine motor, language, social) were reported. An independent t-test was conducted to compare the means, and a linear regression model was used to evaluate the impact of postnatal CMV infection on neurodevelopmental outcomes.

The neurological assessment scores (HINE) were stratified by age groups (3 months, 4–9 months, and 9 months and over) to evaluate the age-specific impact of postnatal CMV infection. Medians for global HINE scores, with 95% CI and the proportion of infants passing the age-specific threshold were reported for each age group.

Log binomial regression was used to report the relative risk of suboptimal hearing outcomes, defined as reduced or absent DPOAEs, and neurological outcomes, categorised by age-specific thresholds using HINE, in postnatal CMV-infected infants compared to uninfected infants. The model accounted for clustering effects and adjusted for for arm, postnatal CMV status, birthweight, sex, age, respiratory infections, vaccination status and breastfeeding frequency. Crude and adjusted risk ratios (RR and ARR) with 95% CI were reported.

All statistical analyses were performed using Stata 18 (StataCorp, 4905 Lakeway Drive, College Station, Texas 77845 USA).

## Inclusivity in global research

Additional information regarding the ethical, cultural, and scientific considerations specific to inclusivity in global research is included in the (S2 Appendix).

## Ethical considerations

The study received approval from the local district health offices and following ethics committees: CURE Children's Hospital of Uganda Research and Ethics Committee (CUREC-2022-41), Uganda National Council of Science & Technology (HS2668ES), and Regional Committee for Medical Research Ethics Western Norway (REK West 256906). Voluntary informed consent was obtained from the parents/caretakers of the infants before participating in the study after an explanation of the nature and purpose of the study, the potential benefits, and risks if

any. The research followed the declaration of Helsinki and good clinical practice guidelines to uphold ethical and governance standards.

## Results

### Study population

The mean age of mothers in the cohort was 24 years (SD: 6.18). Mothers of CMV-infected infants were slightly younger, with 27.9% under 20 years of age compared to 20.2% in the uninfected group. Multiparous mothers accounted for 69.7% of the cohort, although primigravida mothers had a higher proportion of CMV-infected infants (35.3% vs. 25.7%). A history of miscarriages was reported in 25.4% of mothers, with similar distributions between the two groups. HIV-positive mothers were rare (1.7%), and maternal malaria was reported in 1.9% of cases, with no significant differences by CMV status.

Household characteristics showed that 57.1% of households had handwashing facilities, although 32.9% of households reported inconsistent soap availability. Frequent contact with young children was common, reported by 74.5% of mothers. Most households (95.5%) had access to improved drinking water, and 86.5% had improved sanitation. Maternal education levels were generally low, with 64.5% of mothers having only primary education or below. Wealth distribution was similar between the CMV-infected and uninfected groups across the poorest (31.3%), middle (33.4%), and richest (35.3%) tiers (Table 1).

The sex distribution was balanced across the cohort, with 47.9% male and 52.1% female. A higher proportion of male infants was observed in the postnatal CMV-infected group (52.5%) compared to the uninfected group (43.6%). Birthweight was predominantly normal, with 92.5% of infants classified as normal weight. Among postnatal CMV-infected infants, 9.0% were low birthweight compared to 6.0% in the uninfected group. Respiratory diagnoses were reported in 5.7% of infants, with similar rates in both postnatal CMV-infected (5.8%) and uninfected (5.5%) groups. Systemic and infectious diagnoses affected 10.1% of infants, with slightly more in the postnatal CMV-infected group (11.2%) compared to the uninfected group (9.2%). Hospitalisation during the first three months of life occurred in 14.2% of infants, with similar rates in both groups. Medication use was common, with 72.6% of infants receiving medication, and 63.1% of the cohort used antibiotics. Vaccination coverage was high: 78.4% of infants received two doses of the DTP vaccine and 79.0% received two doses of the PCV vaccine. A lower proportion of postnatal CMV-infected infants received both doses compared to uninfected infants (DTP: 75.1% vs 81.5%; PCV: 76.1% vs 81.5%), and this difference was statistically significant (Table 2).

### Developmental assessment findings

The distribution of MDAT DAZ scores showed that 2.1% of the infants had Z scores between -3 and <-2, with a higher rate in the infected group (2.9%) compared to the uninfected group (1.4%). Additionally, 13.0% of the infants had DAZ scores between -2 and <-1, with 15.6% in the infected group and 10.2% in the uninfected group. Most infants had DAZ scores between -1 and <0, and 0 and <1, representing 39.4% and 38.4% of the infected and uninfected infants, respectively. These distributions were similar across both groups (Table 3).

The mean DAZ score for all infants was -0.13 (95% CI: -0.21, 0.05). For the infected group, the mean DAZ score was -0.12 (95% CI: -0.23, -0.02), and for the uninfected group, it was -0.14 (95% CI: -0.24, -0.03). In the overall model analysis, the mean difference was -0.02 (95% CI: -0.17, 0.13) with a p-value of 0.017, indicating no significant difference in neurodevelopmental outcomes between the infected and uninfected infants (Table 4). Additionally, no large

**Table 1. Socio-demographic characteristics of participants.**

| Variables | Total | CMV Uninfected | CMV Infected | CRR (95%CI) | p-value |
|---|---|---|---|---|---|
| | N = 424<br>n (%) | N = 218<br>n (%) | N = 206<br>n (%) | | |
| **Maternal age** | | | | | |
| <20 | 101 (23.9) | 44 (20.2) | 57 (27.9) | 1 | |
| 20–24 | 137 (32.5) | 68 (31.2) | 69 (33.8) | 0.89 (0.68, 1.17) | 0.402 |
| 25–29 | 82 (19.4) | 49 (22.5) | 33 (16.2) | 0.78 (0.56, 1.07) | 0.123 |
| 30 + | 102 (24.2) | 57 (26.1) | 45 (22.1) | 0.76 (0.56, 1.02) | 0.067 |
| **Parity** | | | | | |
| Primigravida | 128 (30.3) | 56 (25.7) | 72 (35.3) | 1 | |
| Multiparity | 294 (69.7) | 162 (74.3) | 132 (64.7) | 0.76 (0.63, 0.93) | 0.009 |
| **History of miscarriages** | | | | | |
| No | 220 (74.6) | 125 (76.7) | 95 (72.0) | 1 | |
| Yes | 75 (25.4) | 38 (23.3) | 37 (28.0) | 1.25 (0.91, 1.72) | 0.165 |
| **Maternal HIV status** | | | | | |
| Negative | 394 (98.3) | 202 (98.1) | 192 (98.5) | 1 | |
| Positive | 7 (1.7) | 4 (1.9) | 3 (1.5) | 1.00 (0.36, 2.75) | 1.000 |
| **Maternal malaria** | | | | | |
| No | 416 (98.1) | 214 (98.2) | 202 (98.1) | 1 | |
| Yes | 8 (1.9) | 4 (1.8) | 4 (1.9) | 0.80 (0.27, 2.34) | 0.680 |
| **Residential Setting** | | | | | |
| Rural | 323 (76.5) | 167 (76.6) | 156 (76.5) | 1 | |
| Peri-urban | 99 (23.5) | 51 (23.4) | 48 (23.5) | 0.93 (0.73, 1.19) | 0.575 |
| **Mother's education level** | | | | | |
| None/Primary | 272 (64.5) | 132 (60.6) | 140 (68.6) | 1 | |
| Secondary/above | 150 (35.5) | 86 (39.4) | 64 (31.4) | 0.84 (0.68, 1.03) | 0.107 |
| **Wealth index tiers** | | | | | |
| Poorest | 131 (31.3) | 63 (29.3) | 68 (33.3) | 1 | |
| Middle | 140 (33.4) | 77 (35.8) | 63 (30.9) | 0.89 (0.68, 1.18) | 0.437 |
| Richest | 148 (35.3) | 75 (34.9) | 73 (35.8) | 0.92 (0.69, 1.21) | 0.546 |
| **Family Meal Sharing Practices** | | | | | |
| No | 112 (26.4) | 56 (25.7) | 56 (27.2) | 1 | |
| Yes | 312 (73.6) | 162 (74.3) | 150 (72.8) | 0.95 (0.76, 1.19) | 0.647 |
| **Frequent contact with young children** | | | | | |
| No | 108 (25.5) | 57 (26.3) | 51 (24.8) | 1 | |
| Yes | 315 (74.5) | 160 (73.7) | 155 (75.2) | 0.92 (0.75, 1.14) | 0.456 |
| **Handwashing facility** | | | | | |
| None | 182 (42.9%) | 102 (46.8%) | 80 (38.8%) | 1 | |
| Available | 242 (57.1%) | 116 (53.2%) | 126 (61.2%) | 1.14 (0.91, 1.43) | 0.252 |
| **Household soap availability for handwashing** | | | | | |
| None | 139 (32.9) | 79 (36.2) | 60 (29.4) | 1 | |
| Sometimes | 157 (37.2) | 73 (33.5) | 84 (41.2) | 1.10 (0.82, 1.49) | 0.492 |
| Always | 126 (29.9) | 66 (30.3) | 60 (29.4) | 0.99 (0.72, 1.35) | 0.940 |
| **Maternal hand hygiene before baby contact** | | | | | |
| Never | 4 (1.0) | 2 (1.0) | 2 (1.1) | 1 | |
| Sometimes | 52 (13.1) | 26 (12.5) | 26 (13.8) | 1.00 (0.33, 3.06) | 1.000 |
| Always | 340 (85.9) | 180 (86.5) | 160 (85.1) | 0.97 (0.36, 2.64) | 0.951 |
| **Improved drinking water** | | | | | |

*(Continued)*

**Table 1.** (Continued)

| Variables | Total | CMV Uninfected | CMV Infected | CRR (95%CI) | p-value |
|---|---|---|---|---|---|
| Unimproved | 19 (4.5) | 10 (4.6) | 9 (4.4) | 1 | |
| Improved | 403 (95.5) | 208 (95.4) | 195 (95.6) | 1.08 (0.63, 1.86) | 0.772 |
| **Improved Sanitation** | | | | | |
| Unimproved | 57 (13.5) | 35 (16.1) | 22 (10.8) | 1 | |
| Improved | 365 (86.5) | 183 (83.9) | 182 (89.2) | 1.16 (0.77, 1.75) | 0.456 |

**Table 2. Infant characteristics.**

| Variables | Total | CMV Uninfected | CMV Infected | RR (95% CI) | p-value |
|---|---|---|---|---|---|
| | N = 424 n (%) | N = 218 n (%) | N = 206 n (%) | | |
| **Sex** | | | | | |
| Male | 202 (47.9) | 95 (43.6) | 107 (52.5) | 1 | |
| Female | 220 (52.1) | 123 (56.4) | 97 (47.5) | 0.94 (0.74, 1.19) | 0.612 |
| **Birthweight** | | | | | |
| Normal | 360 (92.5) | 189 (94.0) | 171 (91.0) | 1 | |
| Low birth weight | 29 (7.5) | 12 (6.0) | 17 (9.0) | 1.07 (0.66, 1.73) | 0.775 |
| **Respiratory diagnoses** | | | | | |
| No | 400 (94.3) | 206 (94.5) | 194 (94.2) | 1 | |
| Yes | 24 (5.7) | 12 (5.5) | 12 (5.8) | 1.00 (0.62, 1.62) | 1.000 |
| **Systemic and infectious diagnoses** | | | | | |
| No | 381 (89.9) | 198 (90.8) | 183 (88.8) | 1 | |
| Yes | 43 (10.1) | 20 (9.2) | 23 (11.2) | 0.96 (0.66, 1.39) | 0.826 |
| **Hospitalisation in the first 3 months of life** | | | | | |
| No | 363 (85.8) | 186 (85.3) | 177 (86.3) | 1 | |
| Yes | 60 (14.2) | 32 (14.7) | 28 (13.7) | 0.92 (0.66, 1.27) | 0.597 |
| **Any medication** | | | | | |
| No | 116 (27.4) | 54 (24.8) | 62 (30.1) | 1 | |
| Yes | 308 (72.6) | 164 (75.2) | 144 (69.9) | 0.88 (0.69, 1.11) | 0.281 |
| **Antibiotics** | | | | | |
| No | 114 (36.9) | 61 (37.0) | 53 (36.8) | 1 | |
| Yes | 195 (63.1) | 104 (63.0) | 91 (63.2) | 0.93 (0.75, 1.16) | 0.524 |
| **Diphtheria-Tetanus-Pertussis vaccine** | | | | | |
| 0 | 8 (1.9) | 2 (0.9) | 6 (3.0) | 1 | |
| 1 | 70 (16.8) | 36 (16.7) | 34 (16.9) | 0.64 (0.40, 1.03) | 0.066 |
| 2 | 327 (78.4) | 176 (81.5) | 151 (75.1) | 0.65 (0.46, 0.92) | 0.016 |
| 3 | 12 (2.9) | 2 (0.9) | 10 (5.0) | 1.00 (0.57, 1.77) | 1.000 |
| **Pneumococcal conjugate vaccine** | | | | | |
| 0 | 8 (1.9) | 2 (0.9) | 6 (3.0) | 1 | |
| 1 | 69 (16.5) | 36 (16.7) | 33 (16.4) | 0.63 (0.39, 1.01) | 0.057 |
| 2 | 329 (79.0) | 176 (81.5) | 153 (76.1) | 0.65 (0.46, 0.92) | 0.016 |
| 3 | 11 (2.6) | 2 (0.9) | 9 (4.5) | 1.00 (0.57, 1.77) | 1.000 |
| **Exclusive breastfeeding** | | | | | |
| No | 6 (1.4) | 3 (1.4) | 3 (1.5) | 1 | |
| Yes | 417 (98.6) | 215 (98.6) | 202 (98.5) | 1.00 (0.25, 4.09) | 0.996 |

**Table 3. Comparison of clinical outcomes between postnatal CMV-infected and postnatal CMV-uninfected groups.**

|  | All infants<br>N = 424<br>n (%) | Postnatal CMV Uninfected<br>N = 218<br>n (%) | Postnatal CMV Infected<br>N = 206<br>n (%) | P-value |
|---|---|---|---|---|
| **Early hearing screening** |  |  |  | **0.001** |
| Normal (Pass) | 318 (75.0) | 179 (82.1) | 139 (67.5) |  |
| Positive (Refer) | 106 (25.0) | 39 (17.9) | 67 (32.5) |  |
| **Neurological Outcomes (HINE)** |  |  |  | **0.769** |
| Above age-specific threshold | 248 (58.5) | 129 (59.2) | 119 (57.8) |  |
| Below age-specific threshold | 176 (41.5) | 89 (40.8) | 87 (42.2) |  |
| **Developmental outcomes (MDAT)** |  |  |  | **0.346** |
| DAZ score of -3 to <-2 | 9 (2.1) | 3 (1.4) | 6 (2.9) |  |
| DAZ score of -2 to <-1 | 55 (13.0) | 34 (15.6) | 21 (10.2) |  |
| DAZ score of -1 to <0 | 167 (39.4) | 81 (37.2) | 86 (41.8) |  |
| DAZ score of 0 to <1 | 163 (38.4) | 83 (38.0) | 80 (38.8) |  |
| DAZ score of 1 to <2 | 30 (7.1) | 17 (7.8) | 13 (6.3) |  |

differences were observed in the MDAT sub-domains (Table 4). Lower mean values were observed with increasing age (S3 Appendix).

## Neurological examination findings

Comparison of postnatal CMV-infected and uninfected infants showed 57.8% of the infected group and 59.2% of the uninfected group scored above the HINE age-specific threshold, while 42.2% of the infected group and 40.8% of the uninfected group scored below the threshold, indicating no significant difference in neurological outcomes between the groups (p = 0.769) (Table 3).

The neurological status assessment was stratified into three age categories: infants aged 3 months, 4 to 9 months, and 9 months and older. For 3-month-old infants, the median global HINE score was 65.5 (q25: 61, q75: 69), with 42.9% achieving a score above the age-specific threshold. At 4 to 9 months, the median score was 71 (q25: 67, q75: 74), with 63.5% passing the threshold. For infants aged 9 months and older, the median score was 74 (q25: 73, q75: 77), with 80.9% passing. These results demonstrate a progressive improvement in neurological scores with age (Table 5).

The regression analysis showed there was no difference in the risk of scoring below the age-specific threshold on the HINE between postnatal CMV-infected and postnatal CMV-uninfected groups (ARR 1.03, 95% CI: 0.78, 1.36; p = 0.809) after adjusting for several variables, including postnatal CMV status, birthweight, sex, age, respiratory infections, vaccination status, breastfeeding frequency, hearing status and trial arm allocation, while accounting for clustering effects (Table 6).

**Table 4. MDAT developmental domain scores by postnatal CMV status in infants.**

| Domains | CMV Uninfected Mean (95% CI) | CMV Infected Mean (95% CI) | Mean difference (95% CI) | Co-efficient | P-value |
|---|---|---|---|---|---|
| **Gross motor** | 0.17 (0.07, 0.27) | 0.07 (-0.03, 0.17) | 0.10 (-0.04, 0.24) | -0.10 (-0.24, 0.04) | 0.164 |
| **Fine motor** | -0.23 (-0.38, -0.08) | -0.31 (-0.48, -0.14) | 0.08 (-0.15, 0.31) | -0.08 (-0.31, 0.15) | 0.485 |
| **Language** | 0.64 (0.54, 0.75) | 0.59 (0.46, 0.72) | 0.06 (-0.11, 0.22) | -0.05 (-0.22, 0.11) | 0.517 |
| **Social** | 0.43 (0.33, 0.53) | 0.47 (0.36, 0.58) | -0.04 (-0.19, 0.11) | 0.04 (-0.11, 0.19) | 0.608 |
| **Full model (overall)** | -0.14 (-0.24, -0.03) | -0.12 (-0.23, -0.01) | -0.02 (-0.17, 0.13) | 0.02 (-0.13, 0.17) | 0.017 |

**Table 5. Age stratified Hammersmith Infant Neurological Examination (HINE) scores.**

| Age | All infants | | | Postnatal CMV-uninfected | | | Postnatal CMV-infected | | |
|---|---|---|---|---|---|---|---|---|---|
| | N | Median (q25, q75) | PASS (%) | N | Median (q25, q75) | PASS (%) | N | Median (q25, q75) | PASS (%) |
| 3 months | 128 | 65.5 (61, 69) | 42.9 | 72 | 66 (59.5, 70) | 44.4 | 56 | 65 (62, 68.5) | 41.1 |
| 4–9 months | 275 | 71 (67, 74) | 63.5 | 136 | 72 (67.5, 74) | 64.7 | 139 | 71 (67, 75) | 63.3 |
| 9 months and over | 21 | 74 (73, 77) | 80.9 | 10 | 74.5 (73, 77) | 90.00 | 11 | 74 (71, 77) | 72.7 |
| Overall (Across 3 age brackets) | 424 | 70 (65,73) | 58.5 | 218 | 70 (64, 73) | 59.2 | 206 | 70 (65, 74) | 57.8 |

## Hearing assessment findings

Among the 424 infants, 318 (75.0%) passed the hearing screening, while 106 (25.0%) had reduced or absent DPOAEs and were referred for further evaluation (Table 3). In the postnatal CMV-infected group, 139 (67.5%) passed the hearing screening, and 67 (32.5%) were referred, whereas in the postnatal CMV-uninfected group, 179 (82.1%) passed, and 39 (17.9%) were referred. The mean difference was significant (p = 0.001), with CMV-infected infants having a higher likelihood of reduced or absent DPOAEs and requiring further hearing evaluation compared to uninfected infants.

At the multivariable level, a log binomial regression analysis showed that postnatal CMV-infected infants had a significantly higher risk of not passing the hearing test, with an adjusted risk ratio (ARR) of 1.99 (95% CI: 1.27, 3.12; p = 0.003). This was after adjusting for postnatal CMV status, birthweight, sex, age, respiratory infections, vaccination status, breastfeeding frequency and trial arm allocation, while accounting for clustering effects (Table 6).

## Discussion

Our study found that 17.8% of infants who were CMV negative at birth screened positive for postnatal CMV at 3 months. The postnatal CMV-infected infants had a higher rate of reduced or absent DPOAEs in the hearing screening compared to uninfected infants, with a with a 1.99-fold increased risk of failed hearing screening among the postnatal CMV-infected group. While some studies have reported no significant association between postnatal CMV

**Table 6. Postnatal CMV infection on hearing and neurological status.**

| Variable | Hearing (Reduced or Absent DPOAE) | | | | HINE (Below age-specific threshold) | | | |
|---|---|---|---|---|---|---|---|---|
| | RR (95%CI) | p-value | ARR (95%CI)[a] | P-value | RR (95%CI) | p-value | ARR (95%CI)[a] | P-value |
| Postnatal CMV Infected | 1.89 (1.24, 2.90) | 0.003 | 1.99 (1.27, 3.12) | 0.003 | 0.95 (0.72, 1.27) | 0.745 | 0.88 (0.67, 1.15) | 0.346 |
| Birthweight | 0.72 (0.43, 1.22) | 0.222 | 0.68 (0.42, 1.1) | 0.117 | 0.84 (0.61, 1.17) | 0.307 | 0.78 (0.55, 1.10) | 0.153 |
| Female | 1.06 (0.70, 1.60) | 0.796 | 1.16 (0.74, 1.82) | 0.521 | 0.85 (0.63, 1.14) | 0.273 | 0.80 (0.60, 1.05) | 0.106 |
| Medication | 0.95 (0.59, 1.51) | 0.820 | 0.88 (0.53, 1.45) | 0.605 | 0.97 (0.72, 1.31) | 0.851 | 0.85 (0.61, 1.18) | 0.329 |
| Age | 0.99 (0.99, 1.00) | 0.001 | 0.99 (0.99, 1.00) | 0.001 | 0.99 (0.98, 1.00) | <0.001 | - | - |
| Respiratory Infections | 1.34 (0.54, 3.31) | 0.524 | 1.82 (0.77, 4.27) | 0.167 | 0.91 (0.48, 1.72) | 0.77 | 1.19 (0.69, 2.05) | 0.526 |
| Vaccination up to 3 months | 0.59 (0.88, 3.90) | 0.581 | 0.55 (0.05, 6.53) | 0.64 | 0.34 (0.05, 2.11) | 0.245 | 0.44 (0.07, 2.75) | 0.377 |
| Breastfeeding frequency | 1.02 (0.96, 1.08) | 0.554 | 1.04 (0.99, 1.08) | 0.052 | 1.01 (0.97, 1.06) | 0.587 | 1.03 (0.99, 1.06) | 0.145 |
| Reduced or Absent DPOAE | - | - | - | - | 1.28 (1.00, 1.62) | 0.048 | 1.20 (0.92, 1.56) | 0.181 |

[a]Adjusted for trial arm allocation.

acquisition and hearing outcomes [30–32], our findings are consistent with studies and case reports demonstrating a link between postnatal CMV infection and adverse hearing outcomes [7,33,34]. While much of the existing research has focused on preterm infants, our study shows these risks are also significant in term infants. However, a failed hearing screen does not necessarily indicate permanent or sensorineural hearing impairment [35,36], as accurate diagnosis requires methods such as automated auditory brainstem response (AABR) testing and extended follow-up periods, which were beyond the scope of our study.

Further assessment of this same follow-up cohort showed no significant differences in neurodevelopmental outcomes between the two groups of term infants, which aligns with multiple prospective studies demonstrating that postnatal CMV infection does not adversely affect neurodevelopment in preterm infants, with no significant negative effects observed within the first few years of life [37–40]. Despite these findings, hearing is essential for developmental progression in children. While hearing can be assessed and definitively diagnosed early, the impact of the resulting developmental challenges and related issues may not become apparent until later in a child's development [41–44]. The short duration of the follow-up period in our study was insufficient to definitively assess the developmental outcomes which would require a longer-term follow-up. Nevertheless, early detection through hearing screening and timely interventions are essential to support healthy development and mitigate potential negative outcomes associated with hearing impairment, including delays in speech and language acquisition, hindered cognitive development, and negative effects on social interactions and academic performance [45–47].

Sub-Saharan Africa accounts for a large share of the global burden of congenital and acquired or early-onset hearing impairment, with incidence rates estimated to be up to three times higher than those in high-resource settings [48–50]. Despite the clear need, the implementation of early childhood hearing screening and intervention programmes has only recently been introduced in a few African countries [51–53]. These pilot screening programmes, modelled on those in high-resource settings, are mainly in the continent's largest economies of Nigeria and South Africa. In the rest of sub-Saharan Africa, despite national health policies recognizing the need for early hearing impairment detection, implementation has been limited due to insufficient healthcare financing [54,55]. African countries with very low Gross Domestic Products (GDPs) face additional challenges, and routine clinical paediatric screenings, including well-baby checks and hearing assessments, along with early intervention programmes, are largely absent from public healthcare [56–58].

In East Africa, findings from large population-based studies in Kenya and Uganda revealed referral rates of 3.6% and 3.7%, respectively, following early infant hearing screenings [59,60]. These studies also demonstrated that screening could be effectively incorporated into routine immunisation clinics by using trained non-specialist health workers [59–61]. A similar strategy was implemented in Nigeria and expanded to involve community health workers, which successfully strengthened the healthcare workforce and proved to be largely successful, at least when conducted on a small scale [51]. Integrating this contextual approach with targeted CMV screening could significantly enhance the detection of both congenital and postnatal CMV, enabling timely and appropriate interventions. As its potential becomes more evident, several countries are exploring this integrated screening method as a strategy for improving early detection and management of CMV-related conditions, highlighting its growing importance in public health efforts [62,63].

## Strengths and limitations

The study's strengths included its use of a matched cohort design, pairing postnatal CMV-infected infants with age-matched uninfected infants, enhancing the validity and reliability of

the group comparisons. The study also employed validated assessment tools, MDAT and HINE, ensuring consistent and reliable measurements that contribute to the robustness of the findings. Lastly, diagnostic confirmation of CMV through PCR testing of saliva and urine samples considered as gold standard practice strengthened the study by reducing the risk of misclassification bias and ensuring accurate identification of postnatal CMV status in the participants. Our study had several limitations including hearing testing being limited to otoacoustic emission testing, which did not allow for diagnostic confirmation of sensorineural hearing impairment due to postnatal CMV. The absence of multiple screenings and the lack of essential diagnostic tools, such as AABR testing and paediatric otoscopes with appropriately sized specula, introduced potential misclassification bias, limiting our ability to classify hearing impairment as either conductive or sensorineural, determine its permanence, and fully understand the nature of hearing impairments in the study population. Additionally, the short follow-up period restricted our ability to assess the long-term neurodevelopmental and hearing impacts of postnatal infection. Additionally, unmeasured potential confounders relating to the risk of hearing impairment and developmental delay may have influenced the results, as they were not controlled for.

Also, the sample size in the follow-up cohort may have been too small to detect subtle but clinically important differences between postnatal CMV-infected and uninfected infants, particularly in neurodevelopmental outcomes. This reduced the study's power and increased the risk of Type II errors. Furthermore, unmeasured confounders related to the risk of hearing impairment may have influenced the results, as these factors were not controlled for in the analysis. Lastly, while the developmental and neurological testing modalities are validated and widely used, they may not provide fully accurate outcomes in very young children due to the evolving nature of infant development. Early infancy assessments can be less reliable, as indicated by our findings showing a trend of improvement with age, which aligns with other evidence suggesting that outcomes may become more reliable as children grow older [64].

## Future directions / recommendations

This sub-study highlights the importance of early hearing detection, with 32.5% of infants in the postnatal CMV group failing the hearing screening. To overcome challenges in resource-limited settings, future efforts should focus on integrating hearing and CMV screening into routine care by leveraging existing health services. Evidence suggests that incorporating hearing screenings into routine immunisation clinics, particularly by using non-specialist and community health workers, can enhance early detection and intervention. Expanding this approach, as demonstrated in successful small-scale implementations, could significantly strengthen public health efforts in similar settings. Additionally, long-term research with definitive assessments is essential to better understand the type, severity, and predictive value of early screenings for hearing and neurodevelopmental outcomes, especially in CMV-infected infants.

The BabyGel trial was not designed to prevent and monitor CMV postnatal transmission, and adjusting for trial allocation showed no effect on the correlation between CMV infection and developmental or hearing outcomes. Further research is needed to better understand the relationship between hygiene practices, postnatal CMV transmission, and their potential impact on hearing impairment. Finally, advocacy for increased government involvement and health system reforms is essential to ensure hearing screening and CMV prevention are prioritized within public health initiatives. Strengthening healthcare infrastructure and expanding screening programs will facilitate early detection and intervention, ultimately reducing the burden of hearing impairment and its associated long-term complications.

## Conclusion

Postnatal CMV infection was associated with more positive hearing screenings, though no significant differences in neurodevelopmental outcomes were observed in early infancy. Exploration into the feasibility of incorporating hearing and CMV screening into routine care will play a vital role in early identification and intervention, improving the management of both hearing and CMV-related conditions in resource-limited settings.

## Supporting information

**S1 Appendix.**
(PDF)

**S2 Appendix.**
(PDF)

**S3 Appendix.**
(PDF)

**S4 Appendix.**
(PDF)

## Acknowledgments

Ministry of Health, Uganda.

## Author Contributions

**Conceptualization:** Noela Regina Akwi Okalany, Thorkild Tylleskär, Kathy Burgoine, Ingunn Marie Stadskleiv Engebretsen.

**Data curation:** Noela Regina Akwi Okalany.

**Formal analysis:** Noela Regina Akwi Okalany, Martin Chebet, Francis Okello.

**Investigation:** Noela Regina Akwi Okalany, Fred Bisso.

**Methodology:** Noela Regina Akwi Okalany, Kathy Burgoine, Ingunn Marie Stadskleiv Engebretsen.

**Project administration:** Noela Regina Akwi Okalany, Andrew D. Weeks, Thorkild Tylleskär, Kathy Burgoine, Ingunn Marie Stadskleiv Engebretsen.

**Resources:** Peter Olupot-Olupot, Andrew D. Weeks, Fred Bisso, Thorkild Tylleskär.

**Supervision:** David Mukunya, Peter Olupot-Olupot, Thorkild Tylleskär, Kathy Burgoine, Ingunn Marie Stadskleiv Engebretsen.

**Validation:** Noela Regina Akwi Okalany.

**Visualization:** Noela Regina Akwi Okalany, Fred Bisso.

**Writing – original draft:** Noela Regina Akwi Okalany, Martin Chebet, Francis Okello, Ingunn Marie Stadskleiv Engebretsen.

**Writing – review & editing:** Noela Regina Akwi Okalany, David Mukunya, Andrew D. Weeks, Thorkild Tylleskär, Kathy Burgoine, Ingunn Marie Stadskleiv Engebretsen.

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
