## [Decision Letter · Decision Letter 0]

12 Nov 2024

PONE-D-24-46089Postnatal cytomegalovirus infection and its effect on hearing and neurodevelopmental outcomes among infants aged 3 – 10 months: a cohort study in Eastern Uganda.PLOS ONE

Dear Dr. Okalany,

Thank you for submitting your manuscript to PLOS ONE. After careful consideration, we feel that it has merit but does not fully meet PLOS ONE’s publication criteria as it currently stands. Therefore, we invite you to submit a revised version of the manuscript that addresses the points raised during the review process.

**ACADEMIC EDITOR: **

**We prefer your study. Please revise according to the advice by reviewers for publications.**

We look forward to receiving your revised manuscript.

Kind regards,

Kazumichi Fujioka

Academic Editor

PLOS ONE

Journal Requirements:

“The European & Developing Countries Clinical Trials Partnership (EDCTP) funded this study.”

Reviewers' comments:

Reviewer's Responses to Questions

**Comments to the Author**

1. Is the manuscript technically sound, and do the data support the conclusions?

Reviewer #1: Yes

Reviewer #2: Yes

Reviewer #3: Partly

2. Has the statistical analysis been performed appropriately and rigorously? 

Reviewer #1: Yes

Reviewer #2: Yes

Reviewer #3: Yes

3. Have the authors made all data underlying the findings in their manuscript fully available?

Reviewer #1: Yes

Reviewer #2: Yes

Reviewer #3: Yes

4. Is the manuscript presented in an intelligible fashion and written in standard English?

Reviewer #1: Yes

Reviewer #2: Yes

Reviewer #3: Yes

5. Review Comments to the Author

Reviewer #1: The manuscript an important clinical condition that is often missed in low-resource settings, however, the authors need to revise the number of study participants whose results they are presenting.

As per the participant enrolment diagram, they report results from 427 of the 438 recruited participants. They lost 10 participants to follow-up and one participant died.

Yet

Parity is reported for 434 participants

Known HIV status is reported for 413 participants

Presence of flu-like symptoms among mothers is reported for 436 participants

Hearing assessment is reported for 424 participants

Comparison of clinical outcomes is reported for 438 participants including those lost to followup and the participant who died.

Reviewer #2: Thank you for the opportunity to review this insightful and valuable study exploring the association between postnatal CMV infection and hearing as well as neurodevelopmental outcomes in infants. The research addresses an important gap in understanding the impacts of postnatal CMV on early childhood health in resource-limited settings, and the use of a community-based cohort is commendable. The study's strengths include the rigorous matching of cases and controls, thorough statistical analysis, and comprehensive assessments for neurodevelopmental and hearing outcomes. I do suggest some minor revisions, which will make the article even more robust and impactful:

- Wording: The use of the term "positive result" in the context of the hearing screening is somewhat confusing, as this may be interpreted as "positive otoacoustic emissions" (i.e. "pass" result), please consider using a different wording, e.g. "failed or abnormal test result"

- The authors mention in the introduction section that "infectious and endemic illnesses, such as HIV/AIDS and malaria, (..) can complicate the immune response in infected individuals and impact viral shedding", but only data on maternal HIV status are reported in the results section. Do the authors have any information on active or dormant malaria cases during pregnancy/lactation in the population?

- Could the authors provide more detailed information on infant feeding practices in both cohorts? (esp. breastfeeding rates?). This sort of information would be very interesting to the readership, especially bearing in mind potential (and likely) routes of viral transmission.

- Figure 1: Study flowchart. I congratulate the authors on the high follow-up rates in this challenging setting! Could the authors please specify the causes of infant mortality? (total no. of 5 babies died).

- Could the authors provide a rationale for screening in saliva and confirmatory testing in urine? (and not vice versa?)

- Reporting of medication use in the first 3 months of life: Did this include over-the-counter medication?

- Do the authors have any information on examination results in infants referred to ENT? Or could they provide a rough estimate of the number of infants actually suffering from hearing loss?

- Out of curiosity: Are the authors planning on conducting a study with longer-term follow-up in the same cohort?

Reviewer #3: Postnatal CMV (pCMV) infection is known to be associated with morbidity in pre-term < 32 weeks or <1500 gram babies. A systematic review reports a possible impact on (late) neurodevelopment in this group but long term hearing loss was not found to be an association (reference 29, Stark AT et al, 2021). Whether pCMV infection is associated with neurodevelopmental compromise or (sensorineural) hearing loss in TERM babies is however less investigated. In general, term babies are believed to not suffer clinical consequences due to their relative maturity.

This is a sub-study that used a cohort recruited for their (ongoing) primary study called the Babygel Study. This Babygel Study is investigating the impact of improving hand hygiene in rural Ugandan households using locally-produced alcohol-based hand gel. The BabyGel study’s primary outcome is the rates of severe infant illness or death in the first 90 days of life. We are not provided details about BabyGel Study but their study protocol is cited (ref 24).

This current (sub) study is a case control study investigating whether or not pCMV infection in TERM babies impact on neurodevelopment or hearing (in the first year of life). The study compared age matched term using validated tools (the Malawi Development Assessment Tool and the Hammersmith Infant Neurological Examination) and hearing ‘loss’ was screened for by Otoacoustic Emission (OAE) testing. Note that absent OAEs could be due to hearing loss (HL) which could be either conductive or sensorineural in origin, wax in the ears, fluid or infection in the middle ear or a malformed inner ear. Formal audiology was not performed.

The study found that neurodevelopment between the two groups were similar, but babies with postnatal CMV failed their hearing screen at a higher rate. The authors appropriately point out that failing a hearing screen does not equate to hearing loss and that a formal hearing assessment is needed to establish actual hearing loss. The study did not provide formal audiological assessments (ABR), so it remains unknown if the babies had hearing loss (conductive or sensorineural). In addition, if formal audiology had been performed, we may have found the hearing loss to be mainly conductive hearing loss, which in young babies is closely related to middle ear infections (otitis media) and not permanent.

Overall, I have concerns about the methodology of this paper and the conclusion. Firstly, this is a sub- group in a study investigating the impact on infection rates by providing hand hygiene to households via a RCT. The randomisation status of participants in the study (so intervention arm (i.e hand hygiene) vs non intervention arm) is not provided nor included in the analysis. Hand hygiene may contribute to less viral infections in the household or baby, which may then result in less middle ear infections, which in turn is associated with less hearing loss. We do not know from this group if there was a predominance of the non-PCMV group in the BabyGel ‘intervention” arm, which may have been a factor in less hearing loss (indirect effect of less viral rep infections).

The Table 1 reporting the parental socio-demographic characteristics and household hygiene should have had a comment about whether there were any statistical differences between the groups (although eyeballing the figures suggests the groups were similar).

The infant data which is really the ‘confounders’ for risks for hearing loss, should have been a separate table and should include factors that predispose to recurrent infections in the household/ child and include whether or not mothers were randomised to “Babygel” , babies’ vaccination status, breast feeding status, the number of respiratory tract infections the babies have had, with a statical significance provided (p values).

Overall, whilst I appreciate you have compared to controls, I am not confident that pCMV contributed to the failed hearing screens as insufficient analysis of risk have been performed and there may have been bias in the groups.

Some specific comments for clarification:

1) Abstract: In the conclusion, what do the authors mean by “Exploration into the feasibility of incorporating hearing and CMV screening into routine care will play a vital role in improving early diagnosis of CMV and hearing impairment in resource-limited settings.’ .The sentence is sweeping

2) Introduction: A brief explanation of BabyGel Study should have been provided in the text. I appreciate you have referenced citation no. 24, but a brief explanation here provides the requisite context. Also. you should clarify what sort of hearing loss you are concerned about. The assumption is you are concerned about sensorineural hearing loss which is permanent, not conductive, which s reversible

3) Methods

1. How were the babies sourced and recruited?

2. Were the babies only assessed by the 5 research assistances? S - no clinical follow up by a medical team (this could be citied as a limitation? )

3. Were babies all breast fed? The assumption is yes in the LMIC setting but this is an important confounder (breast feeding protective against respiratory tract infections and middle ear infections)

4. Reference 27 (for methods) not available as yet in literature (so unable to access)

5. Can you clarify if a ‘failed hearing screen ‘ is based on one screen? was a second screen done ? Failing a hearing screen could be due to a temporary blocked ear that resolves on a second screen

6. Table 1: demographic / epidemiological data: p values would be useful (understand this is a ‘Table 1’, but this is not an RCT and there could be differences between the groups )

7. BabyGel randomisation status of the mothers?

8. Baby data: Should be a separate table. Needs confounders like whether or not these babies had more respiratory infections/ viral upper respiratory tract infections - a risk factor for otitis media, the vaccination status of babies, breast fed, number of children in the household, etc

4) Conclusion. Confusion with ‘public health policy’ need and the aims of this study. I agree that hearing screening is an important public health measure in infancy. However, I found linking the 2 concepts of attempting to link hearing loss with postnatal CMV and the need for a hearing screening because of postnatal CMV confusing and perhaps misleading. The study sends a message that postnatal CMV in term babies can be associated with hearing loss. As outlined above, the there were confounders for hearing loss in term babies not included in the investigations and thus the conclusions are left somewhat questionable

6. PLOS authors have the option to publish the peer review history of their article (what does this mean?). If published, this will include your full peer review and any attached files.

Reviewer #1: **Yes: **Musa Sekikubo, Department of Obstetrics and Gynaecology, School of Medicine, College of Health Sciences, Makerere University, Kampala Uganda

Reviewer #2: No

Reviewer #3: No

---

## [Author Response · Author response to Decision Letter 0]

9 Jan 2025

A. Reviewer 1 comments:

1. Comment 1: The manuscript an important clinical condition that is often missed in low-resource settings, however, the authors need to revise the number of study participants whose results they are presenting. As per the participant enrolment diagram, they report results from 427 of the 438 recruited participants. They lost 10 participants to follow-up and one participant died. Yet: Parity is reported for 434 participants, Known HIV status is reported for 413 participants, Presence of flu-like symptoms among mothers is reported for 436 participants, Hearing assessment is reported for 424 participants, Comparison of clinical outcomes is reported for 438 participants including those lost to follow-up and the participant who died.

Response: Thank you for your comment. The original cohort consisted of 438 participants with confirmed postnatal CMV results. However, the completed outcomes for hearing, MDAT, and HINE assessments were only obtained for 424 participants, which we have used in the final analysis. It is worth noting that some baseline variables have fewer data points due to non-responses by participants, and these missing values have been reflected in the footnotes of the respective tables. The study profile has also been adjusted to reflect this for further clarity.

B. Reviewer 2 comments:

1. Comment 1: Thank you for the opportunity to review this insightful and valuable study exploring the association between postnatal CMV infection and hearing as well as neurodevelopmental outcomes in infants. The research addresses an important gap in understanding the impacts of postnatal CMV on early childhood health in resource-limited settings, and the use of a community-based cohort is commendable. The study's strengths include the rigorous matching of cases and controls, thorough statistical analysis, and comprehensive assessments for neurodevelopmental and hearing outcomes. I do suggest some minor revisions, which will make the article even more robust and impactful: Wording: The use of the term "positive result" in the context of the hearing screening is somewhat confusing, as this may be interpreted as "positive otoacoustic emissions" (i.e. "pass" result), please consider using a different wording, e.g. "failed or abnormal test result"

Response: Thank you for your comment regarding the terminology used in our study. We understand that the term "positive" can have varying interpretations in medical literature. We recognise that these terms can sometimes be ambiguous, and in our review of the literature, we observed that "positive" is used interchangeably in different contexts (e.g. positive vs. normal, positive vs. negative). In our study, a "positive screening test" is defined as DPOAEs being either undetectable or measured below the threshold for normative values in one or both ears, indicating the need for further confirmatory testing This definition is outlined in the Methods section to provide clarity and minimize potential misinterpretation (Lines 155 – 158). By using the term "positive screening test," we aim to emphasize its role as an initial finding that requires additional evaluation, rather than a definitive diagnostic result. We also considered using the term "abnormal," but ultimately decided against it as it may imply a confirmed pathological condition, which is not the intent in a screening context. 

2. Comment 2: The authors mention in the introduction section that "infectious and endemic illnesses, such as HIV/AIDS and malaria, (..) can complicate the immune response in infected individuals and impact viral shedding", but only data on maternal HIV status are reported in the results section. Do the authors have any information on active or dormant malaria cases during pregnancy/lactation in the population?

Response: Thank you for your observation regarding the inclusion of information on endemic illnesses such as malaria in addition to HIV. We have now incorporated data on malaria cases during pregnancy and first three months into results section and Table 1. However, like HIV, the numbers are quite small, limiting our ability to draw meaningful associations or conclusions. 

3. Comment 3: Could the authors provide more detailed information on infant feeding practices in both cohorts? (esp. breastfeeding rates?). This sort of information would be very interesting to the readership, especially bearing in mind potential (and likely) routes of viral transmission.

Response: Thank you for your comment regarding the inclusion of detailed information on infant feeding practices in both cohorts. We agree that this information is of significant interest, particularly given its relevance to potential routes of viral transmission. It is worth noting that breastfeeding is nearly universal in this setting, with approximately 99% of infants breastfed across all weeks up to 12 weeks with a small proportion receiving supplementary feeding, this has been reflected in Table 1. 

4. Comment 4: Figure 1: Study flowchart. I congratulate the authors on the high follow-up rates in this challenging setting! Could the authors please specify the causes of infant mortality? (total no. of 5 babies died). 

Response: Thank you for the positive feedback. We have added the details of the causes of infant mortality to the study profile as requested. 

5. Comment 5: Could the authors provide a rationale for screening in saliva and confirmatory testing in urine? (and not vice versa?)

Response: Thank you for this comment regarding the rationale for screening in saliva and confirmatory testing in urine. The primary reason for this approach is that it aligns with established standards in CMV research. Saliva is widely recognised as an appropriate initial screening sample due to its practicality for large-scale testing, as it is non-invasive, easy to collect in young infants, and has high sensitivity for detecting CMV DNA. Urine has an even higher sensitivity for detecting CMV compared to saliva and is considered the gold standard biological sample for CMV diagnosis due to its typically high viral load and reliability in detecting active infection. This two-pronged approach combines the feasibility and accessibility of saliva for initial screening with the superior accuracy and diagnostic reliability of urine for confirmatory testing. References supporting this rationale have also been added to the manuscript to provide further context and evidence for this methodology (Lines 344 - 348).

6. Comment 6: Reporting of medication use in the first 3 months of life: Did this include over-the-counter medication?

Response: Thank you for this question regarding the reporting of medication use in the first three months of life. In the Ugandan context, many prescribed medications are commonly obtained 'over the counter,' making it challenging to distinguish between traditional over the counter and prescribed medication. In our study, this distinction was not specified during data collection, except for antibiotics, which were previously not reported but have now been included in Table 2. The rationale for including medication use was to assess whether episodes of reduced immunocompetence during the first three months of life increased susceptibility to postnatal CMV acquisition. 

7. Comment 7: Do the authors have any information on examination results in infants referred to ENT? Or could they provide a rough estimate of the number of infants actually suffering from hearing loss? 

Response: Thank you for your comment. Our study did not collect detailed data on confirmed diagnoses, as this was beyond its scope. This was due to the lack of essential diagnostic tools in the ENT department at our facility, including automated auditory brainstem response (AABR) testing and paediatric otoscopes with appropriately sized specula, which necessitated referrals to higher-tier facilities for further evaluation. This limitation in diagnostic capacity restricted our ability to determine the true prevalence and type of hearing loss in the study population. We have acknowledged this as a limitation in the revised manuscript (Lines 349 -355).

8. Comment 8: Out of curiosity: Are the authors planning on conducting a study with longer-term follow-up in the same cohort?

Response: Thank you for your thoughtful question. We agree that it would be highly interesting to conduct a longer-term follow-up in this cohort, particularly as findings in outcomes such as hearing, neurodevelopment, and neurological health often become more apparent and revelatory as children grow older. Unfortunately, the funding for the trial that supported this study has now been completed, which limits our ability to pursue this follow-up at present. We are hopeful that future funding opportunities or collaborations will enable us to build on these findings with extended longitudinal studies.

C. Reviewer 3 comments

1. Comment 1: Postnatal CMV (pCMV) infection is known to be associated with morbidity in pre-term < 32 weeks or <1500 gram babies. A systematic review reports a possible impact on (late) neurodevelopment in this group but long-term hearing loss was not found to be an association (reference 29, Stark AT et al, 2021). Whether pCMV infection is associated with neurodevelopmental compromise or (sensorineural) hearing loss in TERM babies is however less investigated. In general, term babies are believed to not suffer clinical consequences due to their relative maturity. This current (sub) study is a case control study investigating whether or not pCMV infection in TERM babies’ impact on neurodevelopment or hearing (in the first year of life). The study compared age matched term using validated tools (the Malawi Development Assessment Tool and the Hammersmith Infant Neurological Examination) and hearing ‘loss’ was screened for by Otoacoustic Emission (OAE) testing. Note that absent OAEs could be due to hearing loss (HL) which could be either conductive or sensorineural in origin, wax in the ears, fluid or infection in the middle ear or a malformed inner ear. Formal audiology was not performed. The study found that neurodevelopment between the two groups were similar, but babies with postnatal CMV failed their hearing screen at a higher rate. The authors appropriately point out that failing a hearing screen does not equate to hearing loss and that a formal hearing assessment is needed to establish actual hearing loss. The study did not provide formal audiological assessments (ABR), so it remains unknown if the babies had hearing loss (conductive or sensorineural). In addition, if formal audiology had been performed, we may have found the hearing loss to be mainly conductive hearing loss, which in young babies is closely related to middle ear infections (otitis media) and not permanent. Overall, I have concerns about the methodology of this paper and the conclusion. Firstly, this is a sub- group in a study investigating the impact on infection rates by providing hand hygiene to households via a RCT. The randomisation status of participants in the study (so intervention arm (i.e hand hygiene) vs (non-intervention arm) is not provided nor included in the analysis. Hand hygiene may contribute to less viral infections in the household or baby, which may then result in less middle ear infections, which in turn is associated with less hearing loss. We do not know from this group if there was a predominance of the non-PCMV group in the BabyGel ‘intervention” arm, which may have been a factor in less hearing loss (indirect effect of less viral rep infections).

Response: Thank you for your detailed and thoughtful feedback and our responses are as follows:

• Influence of BabyGel trial allocation on hearing outcomes

In this sub-study, our primary focus is on postnatal CMV infection as the exposure variable and its direct association with hearing and neurodevelopmental outcomes. While improved hygiene practices from the intervention arm could theoretically influence hearing outcomes indirectly (e.g., through reduced middle ear infections), this aspect was not the primary aim of our study. We have avoided making broader interpretations regarding the intervention’s overall impact because the main BabyGel trial outcomes are still pending publication. And though not directly described, we incorporated allocation status as a cofactor in the analysis, and it is worth noting that its inclusion did not result in any changes to the findings. We have focused solely on providing context for the sub-study's objectives. Additionally, we have acknowledged the exploratory nature of including allocation data and emphasized the need for further research to better understand the relationship between hygiene practices, postnatal CMV, and related outcomes.

• Hearing screening limitation

We acknowledge that, as you noted, "absent OAEs could be due to hearing loss (HL) which could be either conductive or sensorineural in origin, wax in the ears, fluid or infection in the middle ear or a malformed inner ear," and that formal audiological assessments, such as auditory brainstem response (ABR) testing, were not conducted. Similarly, otoscopy was not performed due to the unavailability of small specula suitable for infants. These limitations, which restricted our ability to determine the type, severity, or definitiveness of hearing impairment among infants who did not pass hearing screening. In the revised manuscript, these limitations have been acknowledged and we have emphasized in the manuscript that the results in this study represent hearing screenings rather than definitive diagnoses. OAE screenings that weren’t passed were described as indicators requiring further diagnostic evaluation, not as conclusive evidence of hearing loss avoid any misrepresentation of the findings. And finally, we have included recommendations for incorporation of formal audiological testing, comprehensive ear examinations, and longitudinal follow-up.

• Methodological rigour and robustness of conclusions

This sub-study was designed to investigate the association between postnatal CMV infection and early hearing and neurodevelopmental outcomes in term infants. The methodology focused on postnatal CMV as the primary exposure variable, using a matched cohort design to ensure comparability between postnatal CMV-infected and uninfected infants. To provide additional context, trial allocation data has been included as a cofactor. The limitations of the study, including the absence of formal diagnostic assessments and the exploratory nature of trial allocation analysis, have been acknowledged in the revised manuscript. These considerations, while important, do not detract from the study’s relevance in addressing its primary aim. This sub-study provides valuable data on postnatal CMV-related outcomes in a resource-limited setting, contributing to the understanding of its potential associations. Future research can expand on these findings with more definitive investigations, including a deeper exploration of the interplay between hygienic practices, postnatal CMV, and related outcomes.

2. Comment 2: The Table 1 reporting the parental socio-demographic characteristics and household hygiene should have had a comment about whether there were any statistical differences between the groups (although eyeballing the figures suggests the groups were similar).

Response: Thank you for your comment. We have added the crude risks and p-values of the respective parental socio-demographic, household/behavioural and infant characteristics in Tables 1 and 2.

3. Comment 3: The infant data which is really the ‘confounders’ for risks for hearing loss, should have been a separate table and should include factors that predispose to recurrent infections in the household/ child and include whether or not mothers were randomised to “Babygel” , babies’ vaccination status, breast feeding status, the number of respiratory tract infections the babies have had, with a statical significance provided (p values).

Response: Thank you for your comment. We have added a separate table to include potential confounders such as household size, vaccination status, breastfeeding frequency, and respiratory infections. The randomisation/ allocation status has been inc

---

## [Editor Report · Decision Letter 1]

21 Jan 2025

Postnatal cytomegalovirus infection and its effect on hearing and neurodevelopmental outcomes among infants aged 3 – 10 months: a cohort study in Eastern Uganda.

PONE-D-24-46089R1

Dear Dr. Okalany,

We’re pleased to inform you that your manuscript has been judged scientifically suitable for publication and will be formally accepted for publication once it meets all outstanding technical requirements.

Kind regards,

Kazumichi Fujioka

Academic Editor

PLOS ONE

Additional Editor Comments (optional):

The revised manuscript is well reflecting the reviewer's advice. And not it is worth for publication.
---

## [Editor Report · Acceptance letter]

24 Jan 2025

PONE-D-24-46089R1 

PLOS ONE

Dear Dr. Okalany, 

I'm pleased to inform you that your manuscript has been deemed suitable for publication in PLOS ONE. Congratulations! Your manuscript is now being handed over to our production team.

Kind regards, 

on behalf of

Dr. Kazumichi Fujioka 

Academic Editor

PLOS ONE
